# Melanoma Vaccines: Comparing Novel Adjuvant Treatments in High-Risk Patients

**DOI:** 10.3390/vaccines13060656

**Published:** 2025-06-19

**Authors:** Joseph C. Broderick, Alexandra M. Adams, Elizabeth L. Barbera, Spencer Van Decar, Guy T. Clifton, George E. Peoples

**Affiliations:** 1Department of Surgery, Brooke Army Medical Center, Fort Sam Houston, San Antonio, TX 78234, USA; elizabeth.l.barbera.mil@health.mil (E.L.B.); spencer.g.vandecar.mil@health.mil (S.V.D.); 2Department of Surgical Oncology, The University of Texas MD Anderson Cancer Center, Houston, TX 77030, USA; amadams2@mdanderson.org; 3Department of Surgical Oncology, Brooke Army Medical Center, Fort Sam Houston, San Antonio, TX 78234, USA; guy.t.clifton.mil@mail.mil; 4LumaBridge and Cancer Vaccine Development Program, San Antonio, TX 78205, USA; gpeoples@lumabridge.com

**Keywords:** melanoma, vaccine, immunotherapy

## Abstract

**Background:** The emergence of checkpoint inhibitors (CPIs) has significantly improved survival outcomes in later-stage melanoma. However, the efficacy of these treatments remains limited, with around 50% of later-stage melanoma patients experiencing recurrence. As variable response rates to CPIs persist, the development of cancer vaccines has emerged as a potential strategy to augment antitumor immune responses. **Results:** This review compares two promising personalized therapeutic cancer vaccine trials in advanced melanoma: Elios Therapeutics’ Tumor Lysate (TL) vaccine and Moderna’s mRNA-4157 vaccine. The TL vaccine, which utilizes yeast cell wall particles (YCWPs) loaded with autologous tumor lysate, and the mRNA-4157 vaccine, which encodes up to 34 patient-specific neoantigens, both aim to stimulate robust tumor-specific immune responses. Both trials were phase 2b randomized studies, with Elios Therapeutics’ trial employing a double-blind, placebo-controlled design, while Moderna’s was open-label. Both trials had roughly equivalent sample sizes (*n* = 187 and *n* = 157, respectively) with similar demographics and disease characteristics. The TL trial reported improvements in disease-free survival (DFS) with a hazard ratio (HR) of 0.52 (*p* < 0.01) over 36 months, whereas the mRNA-4157 trial demonstrated improvements in recurrence-free survival (RFS) with an HR of 0.56 (*p* = 0.053) over 18 months. The TL vaccine exhibited lower rates of related grade 3 adverse events (<1%) compared to the mRNA vaccine (12%). Key differences between the two trials include the use of CPIs, with 100% of patients in the mRNA trial receiving pembrolizumab versus 37% of the patients in the TL trial receiving either an anti-PD-1 or anti-CTLA-4. The production processes also varied significantly, with the mRNA vaccine requiring individualized sequencing and a 9-week production time, while the TL vaccine utilized tumor lysate with a 1–3-day production time. **Conclusions:** While both vaccines demonstrated promising efficacy, future phase 3 trials are needed to further evaluate their potential as adjuvant therapies for melanoma. This review highlights the comparative strengths and limitations of these vaccine platforms, providing insight into the evolving landscape of adjuvant cancer vaccines.

## 1. Introduction

The advent of immunotherapy in the treatment of melanoma has dramatically improved patient prognosis in the past 10–15 years. Specifically, the introduction of checkpoint inhibitors (CPIs) has improved 5-year survival to as high as 50% in patients with metastatic melanoma, whereas previously only ~25% of patients survived a year [1,2,3,4]. Despite these gains, up to half of those receiving checkpoint inhibitor therapy (CPI) may not achieve a significant therapeutic response [3,5,6,7], and they can develop significant toxicity [3,8]. Approximately 50% of patients with stage IIIB/IV melanoma will develop recurrence even after complete resection and adjuvant treatment [9,10,11,12]. Recent treatment advancements have introduced CPIs in the neoadjuvant setting, where it was hypothesized that the more antigenic pre-excision environment would engender a more robust immune response [13]. In this setting, 60% of resectable stage III patients were shown to have a major pathological response; however, 25% of patients were non-responders, and they did poorly even with subsequent surgical and adjuvant treatment [13]. The shortcomings of CPIs are in part due to the absence of an endogenous antitumor response, and several studies have postulated that cancer vaccines may help augment that response [14,15,16].

There are currently two promising trials evaluating the treatment of later-staged (stage III/IV resected) melanoma with cancer vaccines. Elios Therapeutics has developed a tumor lysate, particle-loaded dendritic cell (TLPLDC) vaccine and a tumor lysate, particle only (TLPO) vaccine to induce a tumor-specific immune response in patients with later-stage melanoma. These vaccines load tumor lysate containing the full spectrum of the patient’s antigenic and neoantigenic burden into yeast cell wall particles (YCWPs). The tumor lysate (TL)-loaded YCWPs are used to prime autologous dendritic cells (ex vivo and in vivo, respectively), producing a robust tumor-specific immune response [17,18,19]. Moderna, in their KEYNOTE-942 trial, developed an mRNA vaccine (mRNA-4157) designed to induce an immune response to patient-specific neoantigens. The vaccine is tailored to an individual patient’s mutanome and human leukocyte antigen (HLA) type, encoding up to 34 neoantigens into mRNA, which is endogenously translated and subsequently enters natural cellular antigen processing pathways, producing a robust tumor-specific immune response [12].

Both trials have released phase 1a and phase 1b data demonstrating the safety and efficacy of their respective vaccines in producing tumor-specific immune responses in later-stage melanoma patients [18,20,21]. This review aims to compare the efficacy, safety, and production challenges of the TLPLDC/TLPO and mRNA-4157 vaccines for melanoma treatment in the respective phase II trials. We will analyze key trial design differences, evaluate clinical outcomes, and explore the future implications of these therapies. For the sake of brevity and clarity, Elios Therapeutics’ trial will be referred to as the Tumor Lysate (TL) trial, and Moderna’s trial as the mRNA trial.

## 2. Results

The key components and differences in vaccine and trial design are summarized in Table 1. The TL trial and the mRNA trial both used vaccines that rely on different mechanisms to induce immunity and require different logistics for vaccine production. The mRNA vaccine encodes 34 highly selective neoantigens that are translated in vivo and incorporated into antigen processing pathways, producing a tumor-specific immune response. Producing an mRNA vaccine requires next-generation sequencing of tumor DNA, patient DNA, tumor RNA, and patient HLA typing. The patient-specific data from these processes are provided as inputs to an internal mRNA-4157 bioinformatics system, which then establishes the sequences of up to 34 neoantigens. These neoantigens are incorporated into an mRNA sequence, which is then transferred electronically for manufacturing. The median time from patient enrollment to the first mRNA vaccine injection took 9 weeks. The TL trial uses tumor lysate that contains the entire tumor antigen repertoire, inclusive of neoantigens, and introduces it into natural antigen processing pathways via phagocytosis through autologous dendritic cells (DCs). Tumor lysate is created through freeze/thaw cycling of the tumor sample. The lysate is then loaded into YCWPs, which are made through NaOH/HCL digestion of non-cell wall components and washings, producing B-glycan shells, which are hollow spheres with large pores. For TLPLDC, autologous monocytic cells are isolated from peripheral blood and started toward dendritic cell differentiation via the cytokine milieu. The TL-loaded YCWPs are then phagocytosed by the immature DC, providing the final maturation signal to the DC and the full antigenic repertoire to the cytoplasm for endogenous antigen processing and presentation. For TLPO, the TL-loaded YCWPs are capped with silicate for the retention and attraction of monocytic cells. The TLPO capped particles are injected intradermally for in vivo loading into DC. The TLPLDC vaccine takes 72 h to create, whereas the TLPO vaccine takes 14 h to create. Lot release then requires 2 weeks of sterility testing.

Both trials were prospective, randomized phase 2b trials studying the efficacy of vaccines in patients with stage III and IV (resected) melanoma. A simplified version of each trial’s schema can be found in Figure 1 and Figure 2, adapted from their respective trials [12,19]. The mRNA trial was open-label, with 157 patients with a 2:1 randomization, and its treatment arms were vaccine treatment with pembrolizumab (*n* = 107) versus pembrolizumab monotherapy alone (*n* = 50), designed to detect if the mRNA vaccine would stimulate the immune response and enhance the activity of pembrolizumab. The TL trial, however, was a double-blind trial with 187 patients enrolled in four treatment arms: TLPLDC (*n* = 47), TLPDC + G (tumor lysate, particle-loaded, dendritic cell harvested with granulocyte-colony stimulating factor) (*n* = 56), TLPO (*n* = 43), and placebo (*n* = 41). It was designed instead to detect if the vaccines would stimulate the immune system and, by itself, improve patient outcomes. The trial had two randomization methods, which can be found in Figure 3. Patients were initially randomized 2:1 to either the TLPLDC arm or the placebo arm (*n* = 124) and then later randomized in a 2:1 allocation to the TLPLDC or TLPO arm (*n* = 63). Patients were further divided into the TLPLDC arm based on the use of granulocyte-colony stimulating factor (G-CSF) to enhance leukocyte counts and a smaller phlebotomy versus no G-CSF and a larger volume of blood drawn for DC harvest. This difference was not randomized and instead determined by patient/physician preference. It was later determined that the use of G-CSF produced immature monocytic cells not allowing for complete DC maturation [22]. The results in the G-CSF-produced TLPLDC were identical to the placebo group, which received an injection of matured DCs with empty YCWPs. As a result, the TLPLDC arm without G-CSF had 47 patients, and the total number of patients receiving an active vaccine (TPLDC without G-GSF or TLPO) was 90. Of note, the TL trial was amended to evaluate patients on CPIs once they became the standard of care, with 42% (*n* = 78) of patients having been on CPIs at some point in time (35% prior to trial commencement, 7% during the trial), while the mRNA trial had no significant protocol amendments.

Inclusion criteria for each trial consisted of stage III/IV (specifically stage IIIB for the mRNA trial) melanoma patients capable of being resected and with an Eastern Cooperative Oncology Group performance status of 0–1. Key exclusion criteria included patients who were not disease-free at study entry. The mRNA trial required complete surgical resection with no evidence of disease no more than 13 weeks prior to trial initiation, whereas the TL trial had no time restraints. The trials both had similar age and sex distributions. The trials had slightly different staging representations, with approximately 74% and 26% of patients having stage III and IV melanoma in the TL trial, respectively and 86% and 14% having stage III (C or D) and IV in the mRNA trial, respectively. The TL trial enrolled patients from February 2015 to May 2019, and the mRNA trial enrolled patients from July 2019 to September 2021.

The TL trial reported disease-free survival (DFS) and overall survival (OS) over 36 months, with secondary end points of adverse events also reported. The mRNA trial reported recurrence-free survival (RFS) over 18 months, with secondary end points including distant-metastasis-free survival, safety, and tolerability. The median follow-up time was approximately 36 months for the TL trial and approximately 24 months for the mRNA trial. The DFS and RFS hazard ratios (HRs) were similar in the TL trial (0.52, *p* < 0.001) and the mRNA trial (0.56, *p* = 0.053). Of note, upon subgroup analysis, the TL trial showed that their vaccines had enhanced DFS and OS when used with CPIs in comparison to placebo, but, in the absence of CPIs, the vaccines had a statistically significant increase in OS but not DFS. The TL trial was powered to 80% to detect an HR of 0.5 assuming about 60% recurrence, yielding a sample size of 120 with two-sided a = 0.05, while the mRNA trial required 40 recurrence-free survival events, yielding a sample size of 150 to provide approximately 80% power to detect an HR of 0.5 with one-sided a = 0.10. The TL trial showed statistically similar results between the TLPO and TLPLDC vaccines, as well as the TLPLDC + G vaccine and the placebo. Given the similar clinical outcomes and relative simplicity of producing the TLPO versus the TLPLDC vaccines, future phase III trials will only focus on the TLPO version of the vaccine. In the TL trial, only one patient receiving the vaccine experienced a grade 3 or higher related adverse event, as defined by the National Cancer Institute (NCI) Common Terminology Criteria for Adverse Events (CTCAE)—in this patient, these were symptomatic anemia and hospitalization. In the mRNA trial, 12% of patients receiving the mRNA vaccine experienced grade 3 related adverse events as defined by the CTCAE, the most common of which was fatigue.

## 3. Discussion

The phase IIb trials of TL and mRNA vaccines showed roughly equivalent clinically significant increases in disease-free survival in similar patient populations. There is, however, a significant difference in the complexity of each vaccine’s development. The mRNA vaccine requires an integrated manufacturing process involving next-generation sequencing and proprietary bioinformatics to identify the most immunogenic neoantigens capable of eliciting a tumor-specific immune response. This approach is highly individualized, involving both tumor and blood samples to selectively predict the best neoantigens, making it resource-intensive and costly. In contrast, the TL vaccine has a simpler production process. It uses only a tumor sample to generate the entire antigenic repertoire by producing tumor lysate through freeze–thaw cycles. Additionally, while the median time from patient enrollment to vaccine administration for the mRNA vaccine was approximately 9 weeks, the TL vaccine can be prepared in 1–3 days and released within 2 weeks. Although the mRNA vaccine offers a more targeted approach, its complexity of production may limit its use, particularly in low-resource settings. On the other hand, the TL vaccine, with its shorter, simpler production, could offer a more scalable option if its efficacy is confirmed, but it may require larger volumes of tissue to produce all therapeutic doses of the vaccine regimen.

Further differences lie in how each trial was designed and powered to account for the interpretation of their results. The TL trial, to date, has longer calculated end points of 36 months as opposed to the 18-month end point in the mRNA trial. Both trials were powered to 80%; however, the TL trial had a two-sided a = 0.05, while the mRNA trial had a one-sided a = 0.10. The one-sided α = 0.10 in the mRNA trial increases the likelihood of type I errors (false positives), which could suggest a more favorable result than the two-sided α = 0.05 used in the TL trial, where statistical significance is harder to achieve. The TL trial was also a double-blind, placebo-controlled trial, whereas the mRNA trial was an open-label trial without a placebo arm. The mRNA trial was therefore more prone to observer bias and participant bias, and it had no control for the placebo to fully capture the actual efficacy of its vaccine. The TL trial additionally had a larger sample size; however, the two different randomization schemes produced four treatment groups with relatively smaller patient numbers in comparison to the mRNA trial. This reduced the power of the TL trial to detect meaningful differences within each subgroup and introduced complexity that makes straightforward comparison between treatment arms difficult. The mRNA trial, on the other hand, only had two treatment arms with an otherwise adequately powered study, giving comparably stronger data for its vaccine in combination with pembrolizumab versus pembrolizumab alone.

The significant differences between these trials otherwise lie in CPI use, grade 3 AEs, and vaccine production. CPI therapy has been considered the standard of care in advanced melanoma since its FDA approval in 2014; however, CPIs were not approved for adjuvant use in melanoma until 2017. Every patient in the mRNA trial was on pembrolizumab based on the trial’s design to evaluate if the mRNA vaccine enhanced the efficacy of CPIs. This design reflects current clinical practice, as CPIs are routinely used in the adjuvant setting since their approval. Thus, the mRNA trial provides insight into how the vaccine performs in a setting that closely mimics real-world treatment paradigms. CPIs were not yet approved in the adjuvant setting at the commencement of the TL trial, and its protocol had to be amended to incorporate their use. Upon subgroup analysis, the TL trial showed that their vaccines had enhanced DFS and OS when used with CPIs in comparison to the placebo, but, interestingly, in the absence of CPIs, the vaccines had a statistically significant increase in OS but not DFS. This opens the possibility that the vaccine may have a direct benefit in prolonging survival, even without CPIs, but its ability to prevent recurrence may be more dependent on the additional immune-activating effects of CPI therapy. The mRNA vaccine, in the absence of CPIs, has also shown immunogenicity in other phase 1 studies of resected melanoma [21]. Both of these findings are particularly important because they suggest that the TL and the mRNA vaccine have versatility, potentially benefiting patients both with and without concurrent CPI therapy. However, the subgroup findings from the TL trial should be viewed with caution, as they are exploratory in nature, and the trial was not specifically powered to detect differences in these subgroups. The efficacy of the TL vaccines in combination with CPIs remains a critical area for further investigation, especially given the standardization of CPI therapy in melanoma today. Future trials will need to focus on this combination to provide more definitive evidence of the vaccine’s role in an evolving treatment landscape.

These differences in CPI usage between the two trials also potentially explain the higher incidence of grade 3 AEs in the mRNA trial. Because all patients in the mRNA trial were on pembrolizumab and CPIs are known to cause grade 3 AEs in up to 17% of melanoma patients, it is not surprising that 12% of patients that received the mRNA vaccine in the mRNA trial experienced grade 3 AEs [23]. In contrast, the TL trial included only 78 patients (42%) on CPIs, with only 7% on a concurrent regimen, and it required that patients tolerate CPI therapy for at least 3 months prior to trial enrollment when receiving combined CPIs. This likely contributed to the lower incidence of grade 3 AEs (<1%) reported in the TL trial.

Cancer vaccination has generated considerable excitement as a novel therapeutic approach, though its success has been limited so far. Currently, only three FDA-approved therapeutic cancer vaccines have shown modest results in specific patient populations: TICE, a live vaccine for urothelial carcinoma, sipuleucel-T, a dendritic-cell-based vaccine for prostate cancer, and talimogene laherparepvec (TVEC), an oncolytic viral therapy and in situ vaccine for melanoma [14,24,25,26]. These are just a few examples of the many therapeutic vaccine platforms explored across various cancers. While many of these platforms have shown conceptual promise, most have failed to produce meaningful clinical outcomes [14,15,27]. Melanoma vaccine development, for instance, has seen some success with the oncolytic viral platform (TVEC) but has otherwise faced similar challenges. Previous phase 3 trials with antigen-specific vaccines, such as MAGE-A3 and GM2-KLH, have been unsuccessful [28,29]. The most recent phase 3 trial of a melanoma vaccine, MIND-DC, used adjuvant dendritic cell therapy in stage IIIB/C melanoma vaccines, similarly to the trials discussed in this review, but it also failed to achieve significant results [30]. Developing effective cancer vaccines presents several hurdles, including the low immunogenicity of some cancers, the high disease burden, the complexity of navigating the immunosuppressive tumor microenvironment (TME), and tumor escape mechanisms [15]. However, the TL and mRNA vaccines aim to address some of these challenges by generating a robust, tumor-specific immune response in the adjuvant setting, where the immunosuppressive TME is less established. The TL vaccine uses the entire tumor’s antigenic repertoire loaded into highly immunogenic YCWPs to deliver antigens and innate-stimulating signals to a patient’s immune system to produce an immune response [18]. In contrast, the MIND-DC trial’s dendritic cell vaccine targeted only five known melanoma-specific antigens [30]. The TL vaccine’s approach is less targeted to specific immunogenic antigens and relies on the entire array of tumor proteins, boosted by the immunogenicity of the YCWPs, to allow autologous APCs to generate an effective immune response. The mRNA vaccine selects for highly immunogenic neoantigens and delivers them in concentration to a patient’s immune system. There are theoretical advantages to both approaches, with the TL vaccine generating a broader but potentially more dilute immune response, while the mRNA vaccine generates a highly selective immune response to what are believed to be the most effective antigens.

Other current clinical trials of melanoma vaccines have shown some initial progress but have not yet produced phase II data. A liposomal RNA vaccine (BNT111) and a first-in-class immune-modulatory vaccine (IO102/IO103), in combination with ICIs (immune checkpoint inhibitors), have both shown promising phase 1 data in the treatment of advanced melanoma but have yet to publish phase II results [31,32,33]. Another phase 1 study on a dendritic-cell-based vaccine, WDVAX, also shows promise of significant immune activation [34]. WDVAX is melanoma cell lysate administered with cytokine Granulocyte-Macrophage Colony-Stimulating Factor (GM-CSF) and the innate toll-like receptor 9 agonist CpG oligonucleotide onto a matrix polymer scaffold that allows for precise control of the immunostimulatory agents and stimulation of dendritic cell maturation [34]. Similar to the approach with TLPLDC, this approach may be an alternative method for stimulating a more robust immune response with dendritic-cell-based vaccines. These will be welcome additions to the limited repertoire for advanced melanoma treatment should they prove efficacious in further studies. With the two seemingly effective vaccine approaches presented here that capitalize on unique ways to generate a robust immune response, upcoming phase 3 trials hold the potential to introduce two new vaccine options for melanoma treatment and lay the groundwork for future cancer vaccine platforms.

## 4. Conclusions

These vaccines present promising results for the future of melanoma treatment. The TL vaccine has a simpler, quicker production process, and it contains the entire antigen repertoire, including neoantigens of a given tumor. The mRNA vaccine has a more complex and time-intensive process and focuses on up to 34 highly selective neoantigens to produce a more effective immune response. The TL trial was a prospective, randomized, double-blind, placebo-controlled trial with more long-term data and more stringent thresholds for statistical significance; however, the trial was designed without the inherent inclusion of CPIs and introduced two different randomization schemes with allocation of patients to four different treatment arms. The mRNA trial was a prospective, randomized study, but, at the same time, it was open-label, with no placebo control. It studied the mRNA vaccine in the context of CPI usage, and its associated higher rates of grade 3 AEs can likely be attributed to this difference. With these differences in trial duration, design, and statistical powering, a true comparison of efficacy is difficult to obtain, but the comparative ease of TL’s production is evident. Even with all of the differences in the vaccines and the trials, both studies demonstrated a surprisingly similar significant reduction in recurrences in high-risk melanoma patients. A phase 3 trial is currently underway for the mRNA vaccine, and a similar trial has been approved by the FDA for the TL vaccine. The results of these studies could provide two more adjuvant options to improve the outcome in patients with later-stage melanoma.

## Figures and Tables

**Figure 1 vaccines-13-00656-f001:**
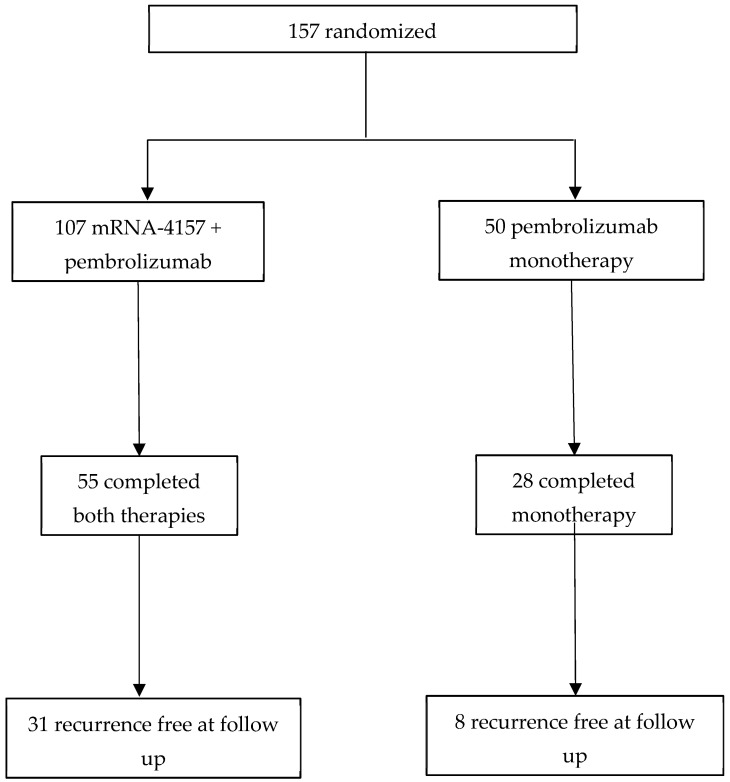
mRNA trial schema. Adapted from Weber et al., 2024 [12].

**Figure 2 vaccines-13-00656-f002:**
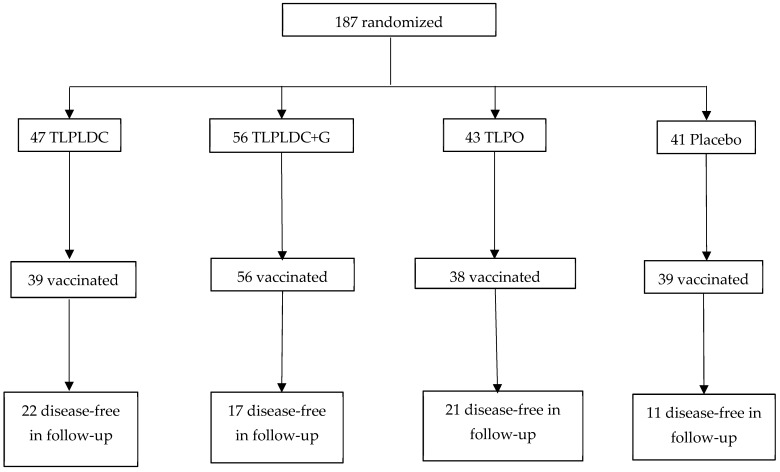
TL trial schema. TL—tumor lysate; TLPLDC—tumor lysate, particle-loaded, dendritic cell; TLPO—tumor lysate, particle only; TLPLDC + G—tumor lysate, particle-loaded, dendritic cell with granulocyte-colony stimulating factor. Adapted from Carpenter et al., 2023 [17].

**Figure 3 vaccines-13-00656-f003:**
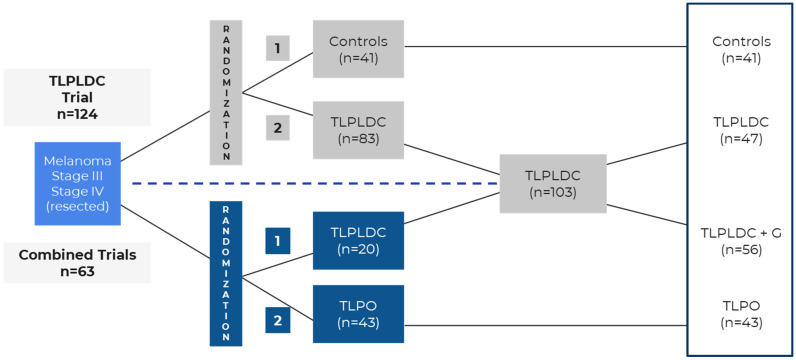
TL trial randomization. TL—tumor lysate; TLPLDC—tumor lysate, particle-loaded, dendritic cell; TLPO—tumor lysate, particle only; TLPLDC + G—tumor lysate, particle-loaded, dendritic cell with granulocyte-colony stimulating factor. Adapted from Carpenter et al., 2023 [17].

**Table 1 vaccines-13-00656-t001:** Trial comparison.

Trial Information			
		Moderna—mRNA *	Elios—TL **
Technology	Vaccine Mechanism	mRNA encodes highly selective neoantigens that are translated in vivo and incorporated into antigen processing pathways, producing a tumor-specific immune response	Tumor lysate contains the entire tumor antigen repertoire loaded in YCWPs introduced into the cytoplasm of autologous dendritic cells (DCs) via phagocytosis
	Samples Required	Tumor DNA and RNA, patient DNA and HLA	Tumor sample only for TLPO Tumor sample and DC for TLPLDC
	Antigen Isolation	Next-generation sequencing of samples is input into an internal mRNA vaccine bioinformatics system, which identifies the ideal neoantigens for mRNA manufacturing	Tumor lysate is created through freeze/thaw cycles of tumor sample
	Antigenic Repertoire	Up to 34 neoantigens	Total repertoire to include all neoantigens
	Time	9 weeks	1–3 days
Trial Design	Phase	2b	2b
	Prospective	Yes	Yes
	Randomized	Yes	Yes
	Blinding	Open label	Double blind
	Placebo Controlled	No	Yes
	Statistical Powering	Required 40 recurrence-free survival events, yielding a sample size of 150 to provide approximately 80% power to detect an HR of 0.5 with one-sided a = 0.10	Assumed 60% recurrence, yielding a sample size of 120 for 80% power to detect an HR of 0.5 with two-sided a = 0.05 Trial was expanded to perform head-to-head comparison of TLPLDC vs. TLPO
	Protocol Amendments	None	Yes—to permit concurrent CPI therapy once FDA approved for use in the adjuvant setting
Patients Enrolled	Melanoma Stage	IIIC or IIID—86%	III—74%
		IV—14%	IV—26%
	ECOG Performance Status	0–1	0–1
	Complete Resection, No Evidence of Disease	Yes, no more than 13 weeks prior to trial enrollment	Yes, no time restriction
Results	Vaccine	mRNA	TLPLDC or TLPO
	Treatment Arms	*n* = 157 (1) mRNA + pembrolizumab (*n* = 107) (2) Pembrolizumab (*n* = 50)	*n* = 187 (1) TLPLDC (*n* = 47) (2) TLPDC + G (*n* = 56) (3) TLPO (*n* = 43) (4) Placebo (*n* = 41)
	Median Follow-Up	~24 months	~36 months
	% of Patients on CPI	100% concurrently (pembrolizumab)	42–35% sequentially, 7% concurrently (pembrolizumab, nivolumab, and/or ipilimumab)
	Recurrence or Disease-Free Survival	RFS: HR = 0.56; *p* = 0.053 over 18 months	DFS: HR = 0.52; *p* < 0.01 over 36 months
	Overall Survival	Not reported	HR = 0.17; *p* = 0.011
	Grade 3 Related AEs (No Grade 4 or 5 Related AEs Reported)	12% of patients	<1% of patients (1 in TLPLDC + G)

TL—tumor lysate; TLPLDC—tumor lysate, particle-loaded, dendritic cell; TLPO—tumor lysate, particle only; ECOG—Eastern Cooperative Oncology Group; CPI—checkpoint inhibitor; TLPLDC + G—tumor lysate, particle-loaded, dendritic cell + granulocyte-colony stimulating factor; RFS—recurrence-free survival; HR—hazard ratio; DFS—disease-free survival; AEs—adverse events; YCWPs—yeast cell wall particles. * Adapted from Weber et al. [12]. ** Adapted from Carpenter et al. [17].

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
