# Peer review of "Melanoma Vaccines: Comparing Novel Adjuvant Treatments in High-Risk Patients"

_vaccines, 2025, doi:10.3390/vaccines13060656_

Round 1

Reviewer 1 Report

Comments and Suggestions for Authors

The review ‘Melanoma Vaccines: Comparing Novel Adjuvant Treatments 2 in High-Risk Patients’ by Joseph Broderick, Alexandra Adams, Elizabeth Barbera, Spencer Van Decar, Guy Clifton and George Peoples is devoted to a detailed comparing of two personalized  therapeutic cancer vaccines  as Elios Therapeutics’ Tumor Lysate  vaccine and Moderna’s mRNA- 4157 vaccine. The manuscript is structured logically and presented clearly. There are no similar reviews at the moment. In the opinion of the reviewer, the presented review can be published in the journal Vaccines without any corrections.

Author Response

Comment 1: In the opinion of the reviewer, the presented review can be published in the journal Vaccines without any corrections.

Response 1: Thank you!

Reviewer 2 Report

Comments and Suggestions for Authors

In the manuscript "Melanoma Vaccines: Comparing Novel Adjuvant Treatments in High-Risk Patients," the authors compared the efficacy, safety, and production challenges between the melanoma vaccines of Elios Therapeutics’ Tumor Lysate (TL) vaccine and Moderna’s mRNA-4157. The TL vaccine is autologous tumor lysate, in the format of particle-loaded dendritic cell (TLPLDC) vaccine or tumor lysate particle only (TLPO). The mRNA-4157 is an mRNA vaccine encodes up to 34 patient-specific neoantigens. The data of both phase 2b clinical trials are from recent reports in PMID:37536936 and PMID:38246194.

Current common melanoma treatments are surgery, checkpoint inhibitors, BRAF targeted therapy, and chemotherapy. Cancer vaccine is a relatively new therapeutic strategy for advanced melanoma patients. The topic of this study is interesting and of clinical importance.

A few minor suggestions:
Table 1 should include references in the table or legend
Some recent related publications (PMID:40215342, PMID:39115419) should be discussed in the Discussion.

Author Response

Comment 1: Table 1 should include references in the table or legend

Response 1: Thank you for pointing this out. This has been corrected. Page 2 line 88.

Comment 2: Some recent related publications (PMID:40215342, PMID:39115419) should be discussed in the Discussion.

Response 2: Thank you for this feedback. I have added a discussion of Gainor et al.’s article into the discussion on p 9 lines 291 – 295. I have also added discussion of Hodi et al.’s article p10 lines 347 – 353. Both of these suggestions fit well into the article.

Reviewer 3 Report

Comments and Suggestions for Authors

The manuscript, titled "Melanoma Vaccines: Comparing Novel Adjuvant Treatments 2 in High-Risk Patients", addresses a very important topic related to recent advances in melanoma vaccine development. The review compares two personalized vaccine models (Elios Therapeutics' tumor lysate (TL) vaccine and Moderna's mRNA-20 4157 vaccine) that are in randomized phase 2b studies. Some minor corrections and further clarification are required before final acceptance:

Minor issues:

Line 56: The authors pose in the sentence ...pathological response..., shouldn't it be ...response against pathology...?. Along the same lines, patents must be replaced by patients.

Line 66: define TL.

Line 83: The authors should incorporate the table title. In table 1, define TLPDC + G

Line 198: The point should be placed after the bibliography 21.

Major issue:

I understand that the authors decided to compare both vaccines due to their personalized strategy and the fact that they are in Phase II studies. However, I believe it is necessary to justify this clearly, as there are other vaccines under study, of which very little is mentioned in the literature.

Author Response

Comment 1: Line 56: The authors pose in the sentence ...pathological response..., shouldn't it be ...response against pathology...?. Along the same lines, patents must be replaced by patients.

Response 1: I appreciate your response. The study cited here uses the term pathological response, not response against pathology, which seems to be in keeping with how other literature describes this result. The latter has been corrected - p 2 line 56.

Comment 2: Line 66: define TL.

Response 2: This has been corrected - p2 line 66.

Comment 3: Line 83: The authors should incorporate the table title. In table 1, define TLPDC + G

Response 3: Line 3: This has been corrected - p 2 line 83.

Comment 4: Line 198: The point should be placed after the bibliography 21.

Response 4: This has been corrected - p 7 line 200.

Comment 5:

Major issue: I understand that the authors decided to compare both vaccines due to their personalized strategy and the fact that they are in Phase II studies. However, I believe it is necessary to justify this clearly, as there are other vaccines under study, of which very little is mentioned in the literature.

Response 5:

Thank you for this feedback. I have added discussion of three other current novel melanoma vaccine trials that have produced promising phase 1 data but have yet to publish phase 2 results, thus highlighting the relative pertinence of the phase 2 trials presented here (p 10 lines 343 – 355). The discussion also contains a summary of previous melanoma vaccine platform attempts, a suggestion for why they were unsuccessful, and then a discussion on how these vaccines are different and therefore deserving of a dedicated review. I hope this addresses your concern adequately and appreciate any further feedback you may have.

Round 2

Reviewer 3 Report

Comments and Suggestions for Authors

I consider the manuscript ready for acceptance. Only two minor issues need to be addressed: define ICI on line 345, and delete an extra point on line 347.

Author Response

Comment: I consider the manuscript ready for acceptance. Only two minor issues need to be addressed: define ICI on line 345, and delete an extra point on line 347.

Response: These have been fixed. Thank you.